# Plasmonic Titanium Nitride Tubes Decorated with Ru Nanoparticles as Photo-Thermal Catalyst for CO_2_ Methanation

**DOI:** 10.3390/molecules27092701

**Published:** 2022-04-22

**Authors:** Diego Mateo, Juan Carlos Navarro, Il Son Khan, Javier Ruiz-Martinez, Jorge Gascon

**Affiliations:** KAUST Catalysis Center (KCC), King Abdullah University of Science and Technology (KAUST), Thuwal 23955, Saudi Arabia; juancarlos.navarrodemiguel@kaust.edu.sa (J.C.N.); ilson.khan@kaust.edu.sa (I.S.K.); javier.ruizmartinez@kaust.edu.sa (J.R.-M.)

**Keywords:** photo-thermal catalysis, carbon dioxide, methanation, titanium nitride, plasmon resonance

## Abstract

Photo-thermal catalysis has recently emerged as a viable strategy to produce solar fuels or chemicals using sunlight. In particular, nanostructures featuring localized surface plasmon resonance (LSPR) hold great promise as photo-thermal catalysts given their ability to convert light into heat. In this regard, traditional plasmonic materials include gold (Au) or silver (Ag), but in the last years, transition metal nitrides have been proposed as a cost-efficient alternative. Herein, we demonstrate that titanium nitride (TiN) tubes derived from the nitridation of TiO_2_ precursor display excellent light absorption properties thanks to their intense LSPR band in the visible–IR regions. Upon deposition of Ru nanoparticles (NPs), Ru-TiN tubes exhibit high activity towards the photo-thermal CO_2_ reduction reaction, achieving remarkable methane (CH_4_) production rates up to 1200 mmol g^−1^ h^−1^. Mechanistic studies suggest that the reaction pathway is dominated by thermal effects thanks to the effective light-to-heat conversion of Ru-TiN tubes. This work will serve as a basis for future research on new plasmonic structures for photo-thermal applications in catalysis.

## 1. Introduction

Photo-thermal catalysis is a sub-discipline of heterogeneous catalysis that combines both the photochemical and thermochemical contributions of light [1,2]. Thanks to the synergy between light and heat, photo-thermal catalysis circumvents some of the intrinsic limitations of pure photochemical processes (i.e., low efficiency values and poor utilization of the low-energy regions of solar spectrum) [3,4,5]. Furthermore, the photo-thermal approach provides localized heat at the active sites, thus avoiding the heating of the whole reactor system and, consequently, increasing the efficiency of the overall process [3,6]. All these features have motivated the rise of photo-thermal catalysis as a promising route to produce fuels and chemicals using sunlight as energy source [7,8,9,10,11,12,13].

In this regard, carbon-based materials, defective semiconductor oxides and metal sulfides have been proposed as good candidates for photo-thermal applications due to their broadband light absorption and efficient light-to-heat conversion [14,15,16]. Metal nanoparticles (NPs) displaying localized surface plasmon resonance (LSPR) are also a sub-family of active photo-thermal structures. Upon illumination under resonant conditions, plasmonic metal NPs generate high-energy hot electrons that emit their excess of energy in the form of heat, thus increasing the local temperature of their surroundings [17,18]. Notwithstanding that a major advantage of plasmonic nanostructures lies in the possibility to tune the position of the resonant wavelength just by adjusting the size or the morphology, metals exhibiting strong LSPR in the visible region typically include costly noble metals such as Au and Ag, thus hampering the wide implementation of this strategy [19,20].

In the last years, transition metal nitrides have been postulated as potential substitutes for noble metals in plasmonic applications owing to their low cost and excellent properties including high electronic mobility, elevated melting point and extended light absorption [21]. Particularly, titanium nitride (TiN) is considered a good aspirant for broadband photo-thermal systems by virtue of its intense LSPR in the visible region, which makes this material a potent sunlight absorber, even exceeding Au and Ag [22]. In fact, recent works have reported the use of hybrid materials containing plasmonic TiN to enhance the photocatalytic activity [23,24]. For instance, the groups of Naldoni and Govorov demonstrated that commercial TiN nanocubes decorated with Pt nanocrystals efficiently decomposed ammonia borane to produce hydrogen (H_2_) thanks to the synergistic effect of hot electrons and photo-thermal effect in TiN [25]. In a similar vein, the use of commercial TiN NPs in combination with indium oxide hydroxide (In_2_O_3−x_(OH)_y_) improved the photocatalytic performance towards the CO_2_ reduction reaction to CO [26]. In this case, plasmonic TiN NPs had a dual positive effect generating hot electrons in combination with heat that boosted the catalytic activity of indium oxide.

Surprisingly, to the best of our knowledge, there are no examples in the literature of the use of TiN structures with controlled morphologies as active plasmonic photo-thermal materials. Herein, we report the preparation of TiN tubes through nitridation of TiO_2_ tubes followed by the deposition of Ru NPs. The as-prepared Ru-TiN photocatalyst exhibits a broad LSPR band across the visible and near-IR and drives the light-mediated CO_2_ methanation reaction at high catalytic rates up to 1200 mmol g^−1^ h^−1^. Mechanistic studies suggest that the reaction pathway is mainly dominated by thermal effects derived from the effective light-to-heat conversion of plasmonic TiN tubes. 

## 2. Results

TiN tubes were obtained by nitridation at 800 °C of precursor TiO_2_ tubes under NH_3_ flow, as previously reported [27]. In a subsequent step, Ru NPs were deposited on TiN tubes using the polyol method, given the simplicity of this procedure to obtain homogeneous distributions of small metal NPs (see Materials and Methods section for further experimental details). X-ray diffraction analysis of the as-prepared TiO_2_ and TiN tubes showed the characteristic diffraction peaks from anatase and TiN phases, thus demonstrating the successful synthetic procedure (Figure 1).

TiN tubes were also characterized by Raman spectroscopy (Appendix A). As it can be seen, TiN tubes exhibited the distinctive Raman peaks at ~260, 406 and 603 cm^−1^ arising from the longitudinal acoustic (LA), second-order acoustic (2A) and transverse optical (TO) modes of TiN, respectively [28]. 

Diffuse-reflectance UV–visible spectroscopy revealed the presence of a broad resonance peak of commercial TiN centered at 425 nm (Figure 2). Interestingly, in the case of TiN tubes, the intensity of the plasmon resonance peak decreased, while its maximum shifted to longer wavelengths (~530 nm). These observations are in line with previous theoretical studies that have demonstrated a red-shift for plasmon resonances in sharp or elongated geometries [29]. Indeed, the morphology is responsible not only for the extended light absorption in the low-energy region of the spectrum, but also for the enhanced heating efficiency, of particular interest for photo-thermal applications [29,30]. In addition to this, the deposition of Ru NPs on TiN tubes increased the light absorption in the visible range. 

X-ray photoelectron spectroscopy (XPS) measurements allowed the study of the physicochemical state of Ti, N and Ru in Ru-TiN samples. Regarding the Ti2p region, we found two secondary components at 455.1 and 456.4 eV corresponding to Ti-N and N-Ti-O together with a major component at 458.7 eV ascribed to TiO_2_ (Appendix A) [31]. These results indicate that the surface of TiN is partially oxidized in the form of oxynitride and anatase phase TiO_2_. This TiO_2_ phase was not detectable with XRD as it basically corresponds to an amorphous layer of TiO_2_ resulting from the spontaneous oxidation of TiN under ambient conditions. Indeed, we also detected additional minor components in XPS related to different oxidation states of Ti (Ti^2+^ and Ti^3+^) that further evidenced the incomplete oxidation of Ti in the sample. In the case of N1s spectra, the main peak can be deconvoluted into one component at 396.9 eV attributed to N bonded to Ti and a second component at lower binding energy originating from N-O bonds in oxynitride TiN_x_O_y_, as previously observed in the Ti2p spectra (Appendix A). The satellite localized at higher binding energies has been ascribed to oxidized species, while some authors have identified the small component at ~396.4 eV as chemisorbed nitrogen [32,33]. Finally, with respect to Ru species, the 3d_5/2_ spectra show two main components of metallic Ru and RuO_2_·xH_2_O at 280.2 and 280.7 eV, respectively (Appendix A). It was also possible to detect signals at 284.5 and 285.0 eV corresponding to metallic Ru 3d_3/2_ and RuO_2_·xH_2_O 3d_3/2_ along with satellites at 282.5 and 286.8 eV [34]. The strong component at 285.1 eV, together with small components at 286.7 and 289.0 eV, are attributed to the C1s spectra of adventitious carbon present in the sample.

The morphology of TiO_2_ and TiN tubes was investigated by Scanning Electron Microscopy (SEM) images (Appendix A). Appendix A shows an overview of the TiO_2_ tubes with a size distribution ranging from 1 to 12 µm. The nitridation step did not significantly alter the morphology and size of TiN tubes, as per Appendix A. Upon higher magnification, SEM images disclosed the presence of TiN tubes with an average diameter of 0.3–1.0 µm and a wall thickness of about 150–300 nm (Figure 3).

We also performed Brunauer–Emmett–Teller (BET) measurements to study the textural properties of both TiO_2_ and TiN tubes (Appendix A). Hence, TiO_2_ tubes displayed a specific surface area of 80 m^2^ g^−1^. After the nitridation process, the surface area of the resulting TiN tubes was decreased to 35 m^2^ g^−1^, a value that is consistent with previous reports [27].

Figure 4 shows High Resolution Transmission Electron Microscopy (HRTEM) images of Ru(2)-TiN tubes together with the particle size distribution. As per these images, Ru NPs were homogeneously distributed on the surface of TiN tubes, with an average particle size of 1.8 ± 0.4 nm. Additional SEM images of Ru-TiN tubes in combination with Energy Dispersive X-ray (EDX) analysis of a selected area allowed the study of the chemical composition of the samples and further demonstrated the presence of Ti, N and Ru (Appendix A).

We evaluated the catalytic activity of Ru(2)-TiN towards the photo-thermal CO_2_ methanation reaction using a quartz photoreactor equipped with a thermocouple and a manometer to monitorize the temperature and pressure, respectively, and a Xe lamp as light source (see Materials and Methods section for further experimental details). Figure 5a shows the temperature and pressure profile of Ru(2)-TiN upon 4 consecutive catalytic cycles. As it can be seen, in all cases, the pressure decreased due to the consumption of reactants and the temperature increased up to 275–300 °C as a consequence of light radiation and the inner exothermicity of the CO_2_ methanation. CO_2_ conversion values after only 6 min of reaction ranged from 86 to 92%, thus demonstrating the excellent activity of the catalyst (Figure 5b). HRTEM analysis of the samples after reaction displayed an average particle size of 2.0 ± 0.5 nm for Ru NPs, a value that slightly differs from the one obtained in fresh samples and excludes the possibility of significant particle sintering or agglomeration (Appendix A). Nevertheless, in an effort to further study the stability of Ru-TiN tubes under reaction conditions, we performed a long-term experiment under continuous flow configuration. As can be seen in Appendix A, the catalytic activity remained stable at CO_2_ conversion values ~50% after 150 min of irradiation; although, after this point, a progressive decrease in the CO_2_ conversion was observed. It should be noted that CH_4_ was the main product, in a similar way to the experiments in batch-type configuration. In order to explore the origin of the catalyst deactivation, we firstly analyzed the spent sample by TEM. Surprisingly, we did not find significant differences in Ru NPs particle size compared to the fresh sample, so we ruled out the possibility of particle sintering as the source of catalyst instability (Appendix A). However, XRD analysis of the spent Ru-TiN tubes showed new diffraction peaks attributable to TiO_2_ phase (Appendix A). Given the fact that Ru-TiO_2_ tubes proved to be much less active towards the CO_2_ methanation reaction (see Table 1), we hypothesize that the catalyst deactivation derives from partial oxidation of TiN tubes under operating conditions.

To have a better insight into the reaction mechanism, we studied the relationship between light intensity and reaction rate. Traditionally, a linear relationship between irradiance and reaction rate is a signature of electron-driven reactions, whereas an exponential relationship indicates that the system is dominated by thermal effects [35]. As depicted in Figure 6a, there is a distinct exponential behavior between the CH_4_ production rate and the power density, therefore suggesting that the thermal contribution is preeminent. Figure 6b shows the temperature profile of Ru(2)-TiN tubes under different light intensities. Noticeably, under the highest irradiance the photocatalyst reached a temperature as high as 287 °C in only 10 min of reaction. Altogether, these results unveil the effective light-to-heat conversion of TiN tubes decorated with Ru NPs and their high catalytic activity towards the CO_2_ methanation reaction.

We performed a series of blank experiments in order to demonstrate the superior catalytic activity of the Ru(2)-TiN photocatalyst (Table 1). To this end, we tested the catalytic performance of bare TiN tubes, Ru NPs supported on TiO_2_ tubes and Ru NPs supported on commercial TiN particles. As can be seen in Table 1, bare TiN tubes produced negligible amounts of CH_4_, thus confirming that Ru NPs are the active sites for the CO_2_ methanation. Ru NPs supported on TiO_2_ tubes displayed a poor catalytic activity compared to the sample based on TiN tubes, probably due to their low absorption and utilization of visible and IR light owing to the wide bandgap of TiO_2_ (~3.2 eV). Additional IR thermal images (Appendix A) of both TiO_2_ and TiN tubes under illumination demonstrated that TiN tubes can produce a twofold temperature enhancement, hence confirming the improved light-to-heat conversion. Indeed, under optimal conditions, Ru NPs supported on TiN tubes exhibited an outstanding CH_4_ production rate exceeding 1200 mmol g^−1^ h^−1^ and a turnover frequency (TOF) of 9.1 s^−1^. It should be mentioned that this catalytic activity is among the highest reported so far for the photo-thermal CO_2_ methanation (Appendix A [10,36,37,38,39,40,41,42]). Conversely, Ru NPs deposited on commercial TiN presented a much lower CH_4_ production rate of 338.7 mmol g^−1^ h^−1^ and a TOF of 2.5 s^−1^. In this case, TiN tubes benefit from an enhanced absorption in the visible and NIR region that boosts the heat generation compared to commercial TiN (Appendix A). These outcomes evidence the excellent photo-thermal attributes of TiN tubes derived from their broadband absorption properties and good light-to-heat transformation. *TOF* was calculated with the reaction rate expressed as CH_4_ produced per mol of *Ru* and per second (*r*), and the ruthenium dispersion (*D_Ru_*), as shown in Equation (1) [43]. The ruthenium dispersion was determined on the basis of a cuboctahedron shape model assuming the average Ru particle size obtained by TEM (Figure 4) [44].
(1)$$TOF\;\left(s^{-1}\right)=\frac{r}{D_{Ru}}$$

In order to prove even more the excellent absorption capabilities of TiN tubes in the visible and IR regions of the spectrum, we compared the catalytic activity for CO_2_ photo-thermal methanation under full spectrum radiation and using a UV cut-off filter (λ > 420 nm) both at constant power density (Appendix A). Remarkably, we found that the catalytic performance in terms of CO_2_ conversion and CH_4_ production rate were similar in both cases, therefore indicating that the absorption of photons from visible and IR is the main driving force for the methanation reaction. Altogether, these results highlight the potential of the Ru(2)-TiN photocatalyst as an efficient material to drive the photo-thermal CO_2_ methanation under concentrated natural sunlight.

## 3. Materials and Methods

### 3.1. Material Synthesis

Titanium nitride (TiN) tubes were prepared following an established method with slight modifications [27]. Briefly, 3 g of TiOSO_4_ (MP Biomedicals, Irvine, CA, USA) was dissolved in a mixture of 12.5 mL of glycerol (Aldrich, ≥99%, St. Louis, MO, USA), 17.5 mL of absolute ethanol (VWR Chemicals, Radnor, PA, USA) and 12.5 mL of diethyl ether (VWR Chemicals). The mixture was stirred vigorously for 5 h and subsequently transferred to a 100 mL stainless steel autoclave. The autoclave was placed in an oven at 140 °C for 12 h. The obtained white precipitate was washed with 1.5 L of absolute ethanol and dried in an oven at 60 °C for 1 h prior to calcination in air at 500 °C for 4 h. After this step, the synthesis yield (mass/mass) was 33 ± 6%. Finally, the obtained TiO_2_ tubes were subjected to nitridation under NH_3_ flow (99.999% purity) for 1 h at 800 °C with a prolonged heating ramp (from room temperature to 300 °C at 5 °C min^−1^, from 300 °C to 700 °C at 2 °C min^−1^ and from 700 °C to 800 °C at 1 °C min^−1^). The nitridation step displayed a yield of 80 ± 10%.

The deposition of Ru NPs on the as-prepared materials was performed using the polyol method. Firstly, 200 mg of TiN tubes and 10.5 mg of RuCl_3_·xH_2_O (Alfa Aesar, Haverhill, MA, USA, Ru content min. 38%) were suspended in 60 mL of ethylene glycol (Aldrich) and ultrasonicated for 30 min. The mixture was transferred to a 200 mL round-bottom flask and heated at 160 °C for 3 h. After this time, the solid was obtained by filtration and subsequently washed with water and acetone. Finally, the obtained Ru(2)-TiN tubes were dried at 70 °C for 2 h. Similar procedure was followed to prepare Ru NPs supported on both TiO_2_ tubes and commercial TiN particles (Aldrich). A total number of 4 replicates were performed in order to ensure the reproducibility of the obtained materials.

### 3.2. Material Characterization

Powder X-ray diffraction patterns of the samples were recorded using a Bruker D8 instrument (Bruker, Billerica, MA, USA) with Cu Kα radiation (*λ* = 1.5418 Å, 40 kV, 40 mA). Diffractograms were acquired over the 2θ range of 10–100°, using a step size of 0.04° and a counting time of 1 s per step. Raman measurements were performed using a confocal Raman microscope WITec Apyron (WITec, Ulm, Germany) equipped with a 532 nm laser and power of 7.0 mW. An integration time of 1 s and accumulation number of 20 were applied in all acquisitions. Diffuse reflectance spectra were recorded using a JASCO V-670 spectrophotometer (JASCO, Tokyo, Japan) with halogen and deuterium lamps as light sources. In order to determine the Ru loading, samples were treated with a mixture of aqua regia and HF, and the supernatant was analyzed using a 5100 ICP-OES instrument (Agilent, Santa Clara, CA, USA). X-ray photoelectron spectra of the samples were analyzed using a Kratos Axis Ultra DLD spectrometer (Kratos Analytical, Manchester, UK) equipped with a monochromatic Al_Kα_ X-ray source (*hν* = 1486.71 eV) operating at 150 W, under ultra-high vacuum (≈10^−9^ mbar). N_2_ adsorption–desorption measurements were made at 77 K using a Micromeritics ASAP 2040 instrument (Micromeritics Instrument Corporation, Norcross, GA, USA). Transmission electron microscopy (TEM) micrographs were obtained with a Titan CT microscope (FEI Company, Hillsboro, OR, USA) operated at an acceleration voltage of 300 kV. For the establishment of the particle size distribution of active phases, close to 200 particles from different micrographs were analyzed. Scanning Electron microscopy (SEM) images of the samples were acquired with a FEI Teneo VS microscope (FEI Company, Hillsboro, OR, USA). The electron beam was accelerated at 3 kV, 25 pA, and the images were acquired at around 10 mm working distance.

### 3.3. Photo-Thermal Tests

Photo-thermal CO_2_ hydrogenation experiments were performed using a quartz reactor (58 mL) equipped with a manometer and a thermocouple in intimate contact with the catalyst surface to monitor the pressure and the temperature, respectively. In a typical experiment, a certain amount of photocatalyst (between 20 and 40 mg depending on the experiment) was uniformly spread on a ceramic crucible covering an area of 1.5 cm × 1 cm. The crucible was placed into the reactor and, after purging the atmosphere with H_2_ gas, the reactor was filled with a mixture of H_2_ and CO_2_ (H_2_:CO_2_ ratio of 4) at a total pressure of 5 bar. A Xe lamp (Peccell Technologies, Yokohama, Japan) was used as light source. For the light intensity experiments, an optical power meter (Thorlabs, Newton, NJ, USA) was used to measure the power density. After reaction, the reactor was directly connected to the inlet of a microGC (SRA Instruments, Cernusco sul Naviglio, Italy) to analyze the gas composition. The gas chromatograph is equipped with 2 columns (Molsieve 5A and PPU) and a TCD detector. Molsieve 5A column analyzes H_2_, CH_4_ and CO and uses Ar as carrier gas. PPU column analyzes CO_2_ and up to C_2+_ hydrocarbons and uses He as carrier gas.

For the flow-type stability reaction, Ru-TiN tubes were loaded into a commercial reaction chamber (Harrick, HVC-MRA-5, Pleasantville, NY, USA). The temperature of the catalyst was probed using one thermocouple located a few millimeters below the catalyst bed. A reaction mixture consisting of H_2_ and CO_2_ (ratio 4:1) was introduced at a total flow of 20 mL min^−1^ and ambient pressure. After purging the atmosphere of the reactor for 10 min, the reactor was irradiated from the top and the gas composition was analyzed every 3 min using a gas chromatograph.

## 4. Conclusions

In summary, we have demonstrated that TiN tubes obtained from the nitridation of TiO_2_ tubes are an excellent photo-thermal material to perform the light-induced methanation of CO_2_ in combination with Ru NPs. TiN tubes display a broad absorption across the visible and near-IR derived from the tubular morphology that facilitates the red-shift of the plasmon resonance peak in TiN. The efficient light-to-heat conversion of TiN tubes boosts the CO_2_ methanation through a thermal-driven pathway, achieving catalytic rates up to 1200 mmol g^−1^ h^−1^. We believe that this work will serve as basis for further research on novel transition metal nitrides with tailored morphologies for improved photo-thermal properties.

## Figures and Tables

**Figure 1 molecules-27-02701-f001:**
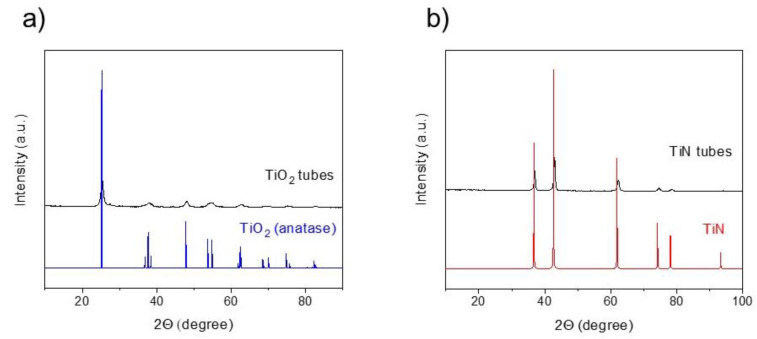
X-ray diffraction (XRD) patterns of (**a**) TiO_2_ tubes and (**b**) TiN tubes.

**Figure 2 molecules-27-02701-f002:**
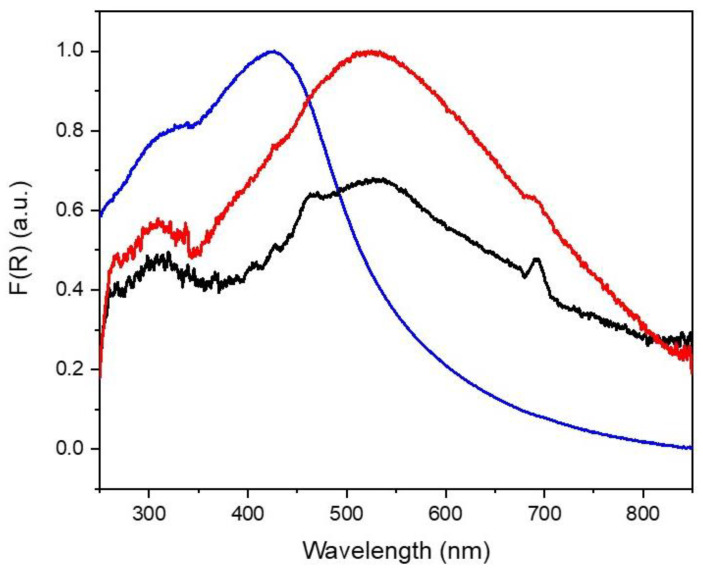
Diffuse-reflectance UV–visible spectra of commercial TiN (blue line), TiN tubes (black line) and Ru(2)-TiN tubes (red line).

**Figure 3 molecules-27-02701-f003:**
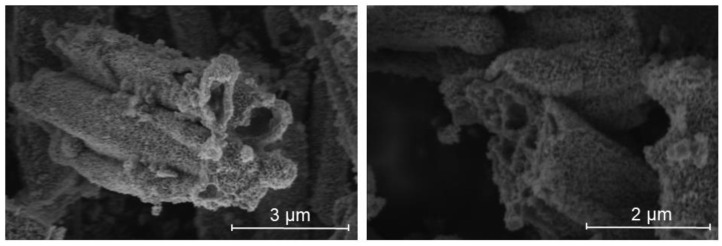
SEM images of TiN tubes.

**Figure 4 molecules-27-02701-f004:**
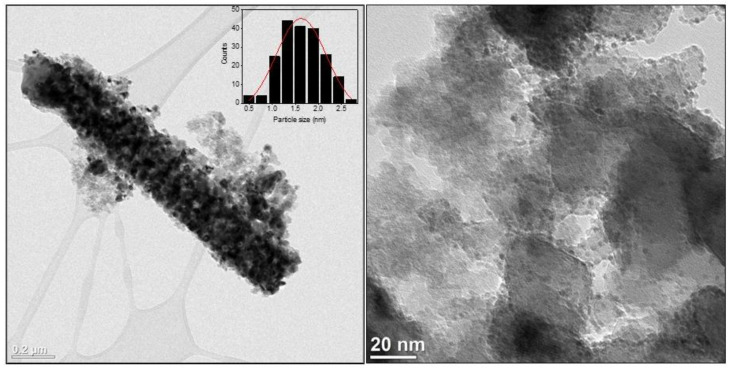
HRTEM images of Ru(2)-TiN tubes before reaction. Inset shows the particle size distribution.

**Figure 5 molecules-27-02701-f005:**
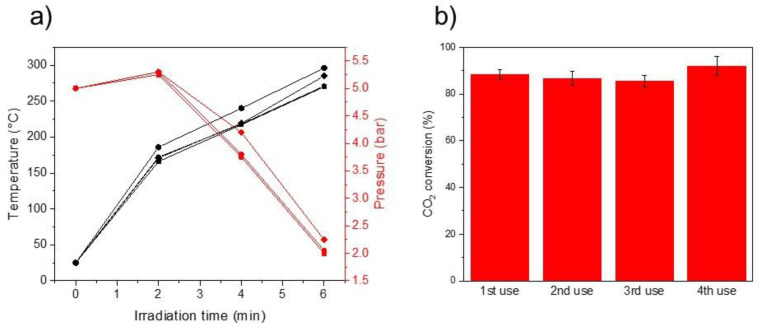
(**a**) Temperature (black line) and pressure (red line) profiles of Ru(2)-TiN photocatalyst under photo-thermal CO_2_ methanation during first (squares), second (circles), third (diamonds) and fourth catalytic cycle (hexagons). (**b**) CO_2_ conversion values upon four consecutive catalytic cycles. Measurements were repeated 3 times.

**Figure 6 molecules-27-02701-f006:**
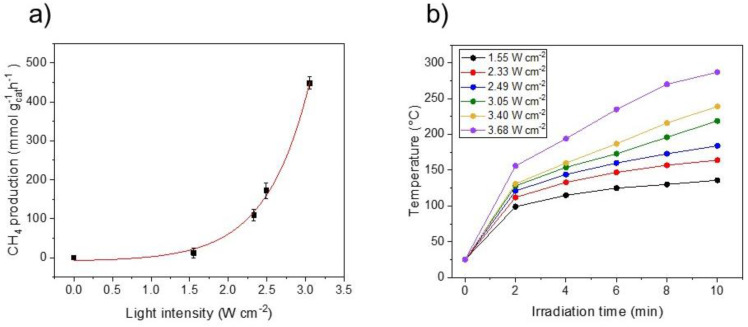
(**a**) Influence of the light intensity on the CH_4_ production rate by Ru(2)−TiN tubes. Red line shows the data trend. (**b**) Temperature profile of Ru(2)−TiN tubes under different light intensities. Measurements were repeated 3 times.

**Table 1 molecules-27-02701-t001:** Metal loading, CH_4_ production rate and turnover frequency (TOF) of different photocatalysts used in this work.

Catalyst	Metal Loading (%)	CH_4_ Production Rate (mmol g^−1^ h^−1^)	TOF (s^−1^)
Ru(2)-TiN tubes	1.7	1215.8	9.1
Ru(2)-TiN commercial	1.6	338.7	2.5
Ru(2)-TiO_2_ tubes	0.7	11.6	-
TiN	-	0.09	-

## Data Availability

Not applicable.

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
