# Peer review of "Plasmonic Titanium Nitride Tubes Decorated with Ru Nanoparticles as Photo-Thermal Catalyst for CO2 Methanation"

_molecules, 2022, doi:10.3390/molecules27092701_

Round 1

Reviewer 1 Report

In this work, Mateo and co-workers reported the development of titanium nitride tubes decorated with Ru nanoparticles as photo-thermal catalyst for CO2 methanation. The authors were successful in the development of such catalysts, which showed relevant performance in catalyzing CO2 methanation. Of note is the result that indicates that this performance is achieved by absorption of visible-IR light, instead of UV light.

The manuscript is well-written and easy to follow, while being scientifically sound. The synthesized materials are relatively well characterized, while providing relevant insights into the photo-thermal catalytic reaction. Furthermore, the manuscript should be of interest for the readership of this journal.

Nevertheless, there are some aspects in this study/paper that must be addressed to in Major Revision:

-What are the synthesis yields (mass/mass, in %) of TiN tubes, Ru(2)-TiN tubes, and TiO2 tubes? The results should be presented in the manuscript as average +/- standard deviation;

-How many replicate synthesis were performed for each material to ensure that they possess reproducible properties? This information should be added to the manuscript;

-What are the band gaps for the obtained TiN tubes and  Ru(2)-TiN tubes?

-How many individual Ru nanoparticles were used to calculate the average particle size by TEM analysis?

-Figures 5 and 6 require error bars, and the indication in the legend of the Figure of the number of replicates; 

-Figure S8 also require error bars, and the indication of the number of replicates;

-The authors should explain the more detail why the results for Ru(2)-TiN tubes and Ru(2)-TiN commercial are so different;

-Did the authors studied more than 4 consecutive catalytic cycles? Did the performance of the catalyst decreased after the fourth assay?

-The authors should include a Table where they compare the performance of their photo-thermal catalyst with other catalysts (even if not photo-thermal ones) for CO2 methanation:

Author Response

Reviewer 1:

In this work, Mateo and co-workers reported the development of titanium nitride tubes decorated with Ru nanoparticles as photo-thermal catalyst for CO2 methanation. The authors were successful in the development of such catalysts, which showed relevant performance in catalyzing CO2 methanation. Of note is the result that indicates that this performance is achieved by absorption of visible-IR light, instead of UV light.

The manuscript is well-written and easy to follow, while being scientifically sound. The synthesized materials are relatively well characterized, while providing relevant insights into the photo-thermal catalytic reaction. Furthermore, the manuscript should be of interest for the readership of this journal.

Nevertheless, there are some aspects in this study/paper that must be addressed to in Major Revision:

We would like to thank Reviewer#1 for their positive comments and for recommending our manuscript for publication in Molecules. Below we elaborate on the shared concerns.

-What are the synthesis yields (mass/mass, in %) of TiN tubes, Ru(2)-TiN tubes, and TiO2 tubes? The results should be presented in the manuscript as average +/- standard deviation;

Following reviewer’s suggestion, we have included the synthesis yield and standard deviation.

-How many replicate synthesis were performed for each material to ensure that they possess reproducible properties? This information should be added to the manuscript;

We thank the reviewer for pointing this out. Indeed, during this work a total number of 4 replicate syntheses were performed in order to ensure a proper reproducibility of the materials. This information has been included in the revised version of the manuscript.

-What are the band gaps for the obtained TiN tubes and  Ru(2)-TiN tubes?

We appreciate reviewer’s comment. Indeed, TiN shows a metallic behavior rather than semiconductor properties, displaying a high conductivity and plasmon resonances both in the visible and NIR regions of the spectrum.

-How many individual Ru nanoparticles were used to calculate the average particle size by TEM analysis?

We thank the reviewer for this comment. As stated in the “Material Characterization” section, close to 200 nanoparticles were measured in order to calculate the average particle size by TEM.

-Figures 5 and 6 require error bars, and the indication in the legend of the Figure of the number of replicates; 

Following reviewer’s suggestion, we have included error bars and the number of replicates in Figures 5 and 6.

-Figure S8 also require error bars, and the indication of the number of replicates;

Following reviewer’s suggestion, we have included error bars and the number of replicates in the new Figure S8.

-The authors should explain the more detail why the results for Ru(2)-TiN tubes and Ru(2)-TiN commercial are so different;

We thank the reviewer for this comment. Ru(2)-TiN tubes take advantage of the enhanced light absorption in the visible and NIR region that ultimately increase the photo-thermal performance. This is obvious from the visible-NIR absorption spectra from both materials (see new Figure S11 in the revised version of the Supplementary Information). The superior absorption of NIR enhances the effective light-to-heat transformation in the case of TiN tubes, therefore increasing the catalytic activity. We have included this information in the revised version of the manuscript.

-Did the authors studied more than 4 consecutive catalytic cycles? Did the performance of the catalyst decreased after the fourth assay?

We appreciate reviewer’s comment. The catalyst did not show any sign of apparent deactivation after 4 consecutive catalytic cycles in a batch reactor. However, in order to further study the stability of the material under reaction conditions, we performed a long-term experiment under continuous flow configuration. As it can be seen in Figure S7 in the revised version of the Supplementary Information, the catalytic activity remained stable after 150 min of continuous irradiation, although after this point a progressive decrease in the catalytic activity was observed. In order to study the origin of this deactivation, we firstly analyzed the spent sample by TEM. Surprisingly, we did not find significant differences in Ru NPs particle size compared to the fresh sample so we ruled out the possibility of particle sintering as the source of catalyst deactivation (Figure S8). However, XRD analysis of the spent Ru-TiN tubes showed new diffraction peaks attributable to TiO2 phase (Figure S9). Given the fact that Ru-TiO2 tubes proved to be much less active towards the CO2 methanation reaction, we hypothesize that the catalyst deactivation derives from the partial oxidation of TiN under reaction conditions. We have included this information in the revised version of the manuscript.

-The authors should include a Table where they compare the performance of their photo-thermal catalyst with other catalysts (even if not photo-thermal ones) for CO2 methanation:

We appreciate reviewer’s comment. A comparison table of different photo-thermal catalysts for CO2 methanation is presented in the Supplementary Information.

Reviewer 2 Report

The paper describes the synthesis of TiN nanomaterials for a photochemical reaction. The results demonstrated a higher absorption over visible and near-IR region which is in line with other literature data. Detailed analysis has been performed by XPS, XRD and TEM measurements. The catalytic activity was reported to be higher as compared to some model catalysts.

The results are interesting and would justify the publication provided that the following comments to be addressed.

  1. Could you provide the C-balance from the product analysis? There could be some other products like C and CH3OH that cannot be detected by GC analysis.

  1. Could you elaborate more on nitridation step. What was the purity of NH3 gas and all other chemicals?

  1. Explain why the temperature and light intensity were changed at the same time (Figure 6b)? Usually to get a proper dependence only one parameter should be changed at a time while the other should be kept constant. Therefore, it is necessary to show the conversion as a function of time at a constant temperature. If the temperature increases due to reaction heat, then you probably run the reaction under heat transfer limited conditions and kinetic data (reacton rates) cannot be seen as reliable. You would need to check this and redo the experiments under forced cooling of your reactor otherwise the data have no scientific meaning.

  1. Explain why strong Ti-O bands were observed in XPS but no TiO2 phases were present in XRD? In fact, the intensity of Ti-O bands exceeds that of Ti-N bands so the material seems to be a mixture of different phases with a high content of TiO2.

  1. Figure S5. Make parts a,b,c. Remove EDX plot – it does not provide any useful information.

  1. Figure S5. Why the colour of Ru is uniform across the entire area? Or this is just background?  

Author Response

Reviewer 2

The paper describes the synthesis of TiN nanomaterials for a photochemical reaction. The results demonstrated a higher absorption over visible and near-IR region which is in line with other literature data. Detailed analysis has been performed by XPS, XRD and TEM measurements. The catalytic activity was reported to be higher as compared to some model catalysts.

The results are interesting and would justify the publication provided that the following comments to be addressed.

  1. Could you provide the C-balance from the product analysis? There could be some other products like C and CH3OH that cannot be detected by GC analysis.

We thank the reviewer for this comment. The carbon balance was >90 % for all the photo-thermal experiments. CO, ethane or other hydrocarbons were below the detection limit of our gas chromatograph, being CH4 the main product.

  1. Could you elaborate more on nitridation step. What was the purity of NH3 gas and all other chemicals?

We thank the reviewer for pointing this out. Following reviewer’s suggestion, we have included the purity of NH3 and all other chemicals used in this work in the revised version of the manuscript.

  1. Explain why the temperature and light intensity were changed at the same time (Figure 6b)? Usually to get a proper dependence only one parameter should be changed at a time while the other should be kept constant. Therefore, it is necessary to show the conversion as a function of time at a constant temperature. If the temperature increases due to reaction heat, then you probably run the reaction under heat transfer limited conditions and kinetic data (reacton rates) cannot be seen as reliable. You would need to check this and redo the experiments under forced cooling of your reactor otherwise the data have no scientific meaning.

We appreciate the comment. The reason why temperature changes with light intensity is because it is actually the light-to-heat conversion the responsible of the increase in the temperature. Under our experimental conditions, both parameters are interconnected as the higher the light intensity, the higher the temperature displayed by the catalyst. It is true that some photo-thermal systems make use of external heating sources, but this is not our case. We also agree with the reviewer when pointing out that changing one of the parameters can help to determine the role of both light and heat. For instance, when the temperature is constant and one varies the light intensity, typically there is a linear relationship between light intensity and the catalytic rate. In the case of keeping constant light intensity and increasing the reaction temperature by external heating, an exponential relationship will be obtained between the rate and the temperature.

In our case, we observed an exponential relationship between the reaction rate and the light intensity, so demonstrating that it is the effective light-to-heat conversion the origin of the catalytic activity. These results indicate that the reaction mechanism is predominantly ruled by thermal effects rather than non-thermal effects. Obviously these non-thermal effects are present, but in our case they represent a minor contribution compared to thermal effects. Another piece of evidence to show that the mechanism is mostly ruled by an effective light-to-heat conversion is the absence of activity using UV light compared to the superior activity under visible and infrared light. Furthermore, blank experiments at room temperature led to negligible CO2 conversion values.

  1. Explain why strong Ti-O bands were observed in XPS but no TiO2 phases were present in XRD? In fact, the intensity of Ti-O bands exceeds that of Ti-N bands so the material seems to be a mixture of different phases with a high content of TiO2.

We appreciate reviewer’s comment. This mismatch between XRD and XPS has been previously reported in TiN materials and the reason lies in the spontaneous oxidation of the outermost layer of TiN. This thin layer of amorphous TiO2 remains undetectable for XRD but is probed by XPS measurements (indeed, XPS is a surface technique with a short penetration depth, hence the large contribution of Ti-O bands). 

  1. Figure S5. Make parts a,b,c. Remove EDX plot – it does not provide any useful information.

Following reviewer’s suggestion, we have labelled the different pictures with letters and removed the EDX plot.

  1. Figure S5. Why the colour of Ru is uniform across the entire area? Or this is just background?

We thank the reviewer for pointing this out. The colored dots corresponding to Ru were barely distinguishable in the first version of Supplementary Information, so we have improved this figure in the revised version using a color with more contrast.

Reviewer 3 Report

The topic of the manuscript is  interesting, however some points need to be discussed in more details and some information need to be complemented. Therefore, in my opinion, the manuscript, as it is, is not suitable for publication. Mayor revisions are needed before new submission.

Based on the results provided by the authors I would define its system as a Ru -TiO2@TiN hybrid material rather than Ru@TiN. I came to this conclusion combining XPS, which is a surface sensitive technique with XRD, which is a bulk technique. Based on XPS Ti-O seems to be the most intense component in the Ti2p spectra, while in the XRD pattern TiN is the only detectable phase. If I am correct and considering that the case, the role of TiO2 in the ternary system need to be carefully discussed. In this direction, a plausible reaction path considering the electron interband and/or intraband transition between the three components would be highly useful in order to understand the enhanced catalytic behaviour reported by the authors. On the other hand, comparison with state of the art catalysts need to be included, in order to get an idea about the novelty of the results.

Beside these points, other less relevant aspects need to be considered.

  1. Blank experiment in absence of light is necessary
  2. Ru metal loading is missed, as well as spectroscopic characterization of the commercial Ru/TiN and Ru/TiO2 systems.
  3. XPS data need to be revised. The authors need to check the FWHM, which should be the same in both Ti2p3/2 and Ti2p1/2, and Ru3d5/2 and Ru3d3/2 core levels. Also I am questioning about the reason for the different line shape used in the RuO2.xH2O component compared to the Gaussian shape employed for the rest of elements.
  4. Experimental section need to be improved.
  5. The Ru particle size in the study is 1.8-2nm. In this direction, the authors need to discuss the role of the particle size on the catalytic performance. And based on that justify why a polyol method has been used in the synthesis.

Author Response

Reviewer 3

The topic of the manuscript is  interesting, however some points need to be discussed in more details and some information need to be complemented. Therefore, in my opinion, the manuscript, as it is, is not suitable for publication. Mayor revisions are needed before new submission.

Based on the results provided by the authors I would define its system as a Ru -TiO2@TiN hybrid material rather than Ru@TiN. I came to this conclusion combining XPS, which is a surface sensitive technique with XRD, which is a bulk technique. Based on XPS Ti-O seems to be the most intense component in the Ti2p spectra, while in the XRD pattern TiN is the only detectable phase. If I am correct and considering that the case, the role of TiO2 in the ternary system need to be carefully discussed. In this direction, a plausible reaction path considering the electron interband and/or intraband transition between the three components would be highly useful in order to understand the enhanced catalytic behaviour reported by the authors. On the other hand, comparison with state of the art catalysts need to be included, in order to get an idea about the novelty of the results.

We appreciate reviewer’s comments. Regarding the possible role of the layer of TiO2, we consider that, in the best-case scenario, it would be negligible compared to the role of TiN as photo-thermal support. The reason is the predominant thermal contribution on the reaction mechanism, evidenced by the exponential relationship between the light intensity and the catalytic rate. In addition to this, experiments using a UV cut off filter (λ>420 nm) showed comparable results to the ones obtained under full spectrum, thus evidencing that the UV light has a little to none effect on the catalytic activity. Therefore, this observation excludes a significant role of TiO2 in the reaction mechanism. Lastly, additional experiments in the revised version of the manuscript (Figure S7 and S9) have proved that the partial oxidation of the TiN tubes under reaction conditions has a negative effect on the catalytic activity, so indicating that TiO2 may show even an adverse effect. Following reviewer’s suggestion, we have included a comparison table of different photo-thermal catalysts for CO2 methanation in the Supplementary Information.

Beside these points, other less relevant aspects need to be considered.

  1. Blank experiment in absence of light is necessary

Following reviewer’s suggestion, we have performed a blank experiment in the absence of light. Under these conditions, it was not possible to detect any product.

  1. Ru metal loading is missed, as well as spectroscopic characterization of the commercial Ru/TiN and Ru/TiO2 systems.

We appreciate reviewer’s comment. Regarding the Ru metal loading, it can be found in Table 1 in the main manuscript. Following reviewer’s suggestion, XPS characterization was carried out for the mentioned systems and the results are presented in the Supplementary Information (Figure S13). Considering the sample Ru/TiO2, the principal oxidation state is +4 (458.2 eV), and it is observed at 459.7 eV a small peak related to the Ru 3p. Considering the Ru, only Ru metallic is observed at 280.1 eV (Ru 3d5/2), 284.3 eV (Ru 3d3/2) and 282.5 eV (satellite). For the commercial Ru/TiN, the results observed are different to the expected. The major compound observed in the Ti2p region is TiO2 (458.6 eV), and two minor compounds at 456.3 eV and 454.9 eV, attributed to N-Ti-O and Ti-N respectively. As it was commented in the manuscript, being TiO2 the principal Ti species observed on the XPS indicates an oxidation of the surface. In addition to the different Ti contributions observed in this region, two possible Ru species are observed at 461.3 eV (Ru metallic) and 462.7 eV (RuO2·xH2O). These species have been observed too in the region Ru 3d at 280.1 eV and 284.3 eV for metallic ruthenium and 281.1 eV and 285.5 eV for hydrated ruthenium, and the satellites at 282.3 eV and 286.1 eV.

  1. XPS data need to be revised. The authors need to check the FWHM, which should be the same in both Ti2p3/2 and Ti2p1/2, and Ru3d5/2 and Ru3d3/2 core levels. Also I am questioning about the reason for the different line shape used in the RuO2.xH2O component compared to the Gaussian shape employed for the rest of elements.

Following reviewer´s comment, we have revised the XPS results. Both Ti2p3/2 and Ti2p1/2, and Ru 3d5/2 and Ru3d3/2 were fixed in the final manuscript, and additional deconvolution´s peaks were added to the Ti2p due to the presence of different oxidation states for titanium. In the case of the different line shape employed on Ru species, it was based on the work presented in Surface and Interface Analysis from D. J. Morgan, which was mentioned in the main manuscript.

  1. Experimental section need to be improved.

Following reviewer’s suggestion, we have improved the experimental section by providing more details on material synthesis yields, number of replicate synthesis and purity of reagents.

  1. The Ru particle size in the study is 1.8-2nm. In this direction, the authors need to discuss the role of the particle size on the catalytic performance. And based on that justify why a polyol method has been used in the synthesis.

We thank the reviewer for pointing this out. We completely agree with the reviewer when highlighting the role of particle size on the catalytic performance. The scope of this work is not, however, a comprehensive evaluation of the effect of the particle size and other experimental parameters like the metal loading on the catalytic activity. Our goal was to demonstrate the potential of TiN tubes as an active photothermal material owing to their capability to transform light into heat. The polyol method has been described as an easy strategy to prepare nanoparticles under moderate reaction conditions, so we made this choice in order to deposit Ru nanoparticles on the surface of our TiN tubes.

Reviewer 4 Report

The manuscript describes the development of plasmonic titanium nitride tubes coupled with Ru nanoparticles applied as photothermal catalysts for CO2 reduction in the presence of hydrogen. Basic characterization techniques were used. It seems to me that the manuscript was prepared in a rush. Usually, the thermal effect is controlled with a heater and not with light irradiation. There are some important issues that are not clear. Where was the thermocouple placed in the reactor? Is the temperature measured refer to the surface of the catalyst film? Was the temperature stable? Apparently it was not stable during the course of the reaction. Therefore, there is no clear correlation of the effect of temperature on the efficiency. Also, there is no experimental evidences for the pure photocatalytic conversion of CO2. Based on these, I suggest rejection of the submission.

Author Response

Reviewer 4

The manuscript describes the development of plasmonic titanium nitride tubes coupled with Ru nanoparticles applied as photothermal catalysts for CO2 reduction in the presence of hydrogen. Basic characterization techniques were used. It seems to me that the manuscript was prepared in a rush. Usually, the thermal effect is controlled with a heater and not with light irradiation. There are some important issues that are not clear. Where was the thermocouple placed in the reactor? Is the temperature measured refer to the surface of the catalyst film? Was the temperature stable? Apparently it was not stable during the course of the reaction. Therefore, there is no clear correlation of the effect of temperature on the efficiency. Also, there is no experimental evidences for the pure photocatalytic conversion of CO2. Based on these, I suggest rejection of the submission.

We appreciate reviewer’s comments. Although we agree with the reviewer’s statement about the possibility to control temperature by external heating, in our system we do not use any external source to increase the temperature of the catalyst, but we take advantage of the ability of TiN tubes to transform light into heat. Both cases, irrespective of the source of heat, meet the requirements for photo-thermal catalysis and there are many examples of the latter approach in the literature.

Regarding the question about the position of the thermocouple, it was placed inside the reactor and in intimate contact with the surface of the catalyst. We have clarified this issue in the revised version of the manuscript.  

As explained above, we do not use an external heater to establish a temperature set point, but it is the inner photo-thermal conversion of the catalyst which dictates the maximum temperature displayed by the catalyst. According to our data, the higher the light intensity, the higher the temperature and therefore the higher the catalytic activity. We respectfully disagree with the reviewer when stating that there is no experimental evidence for pure photocatalytic conversion of CO2, as the light intensity experiments clearly indicate that the reaction mechanism is mainly driven by thermal enhancement (see Figure 6a). Furthermore, blank experiments at room temperature led to negligible CO2 conversion values. These results reveal that in this case the non-thermal contribution to the catalytic activity is negligible, while reaction mechanism is dominated by thermal effects derived from the light absorption of TiN tubes.

Round 2

Reviewer 1 Report

The authors have addressed my comments, and so, my recommendation is for acceptance.

Author Response

We would like to thank Reviewer#1 for their positive comments and for recommending our manuscript for publication in Molecules.

Reviewer 3 Report

The manuscript has  improved according to the reviewer comments, but still some question remains open:

1) the catalyst amount used in the catalytic studies need to be adressed

2) the reason why a particle size of 1.8nm has been selected. I understand the argument given by the authors that particle size effects are not the scope of the work, but the reader may question why such a small particle size has been selected? A comment need to be provided by the authors.  

3) Table S1 need to be completed, including variables such as the temperature, gas composition, pressure, .... 

Author Response

Reviewer 3:

The manuscript has  improved according to the reviewer comments, but still some question remains open:

We would like to thank Reviewer#3 for their positive comments and for recommending our manuscript for publication in Molecules. Below we elaborate on the shared concerns.

1) the catalyst amount used in the catalytic studies need to be adressed

We appreciate reviewer’s suggestion. We have included the information regarding catalyst amount in the revised version of the manuscript (see Materials and Methods section).

2) the reason why a particle size of 1.8nm has been selected. I understand the argument given by the authors that particle size effects are not the scope of the work, but the reader may question why such a small particle size has been selected? A comment need to be provided by the authors.  

We appreciate reviewer’s concern. Indeed the reason to choose this procedure was more due to the simplicity to obtain homogeneous distribution of metal NPs rather than looking for a certain particle size. We have included a note on this issue in the revised version of the manuscript.

3) Table S1 need to be completed, including variables such as the temperature, gas composition, pressure, ...

Following reviewer’s suggestion, we have included experimental details on temperature, gas composition and pressure in the revised version of Table S1.

Reviewer 4 Report

Figure 6a presentes the activity vs. the intensity of the light used. Not the effect of the temperature are the authors mentioned in their responce. There is no data shown the pure photocatalytic effect on activity. Therefore, I cannot recomment acceptance

Author Response

Figure 6a presentes the activity vs. the intensity of the light used. Not the effect of the temperature are the authors mentioned in their responce. There is no data shown the pure photocatalytic effect on activity. Therefore, I cannot recomment acceptance

We thank the reviewer for their comment. The goal of Figure 6a is not to present the effect of temperature, but the effect of the light intensity on the catalytic rate. This is a very straightforward methodology to study if the system is dominated by thermal or non-thermal effects. Again, we would like to emphasize that the pure photocatalytic effect has been ruled out according to the exponential relationship between light intensity and catalytic rate. This indicates that the thermal enhancement is dominant, while the pure photocatalytic contribution is negligible.